An approach to the dermatological classification of histopathological skin images using a hybridized CNN-DenseNet model

De Anubhav 1
Mishra Nilamadhab 1
Chang Hsien-Tsung smallpig@widelab.org 2 3 4 5
1 School of Computing Science & Engineering, VIT Bhopal University , Madhya Pradesh , India
2 Department of Computer Science and Information Engineering, Chang Gung University , Taoyuan , Taiwan
3 Department of Physical Medicine and Rehabilitation, Chang Gung Memorial Hospital , Taoyuan , Taiwan
4 Artificial Intelligence Research Center, Chang Gung University , Taoyuan , Taiwan
5 Bachelor Program in Artificial Intelligence, Chang Gung University , Taoyuan , Taiwan
Chaki Jyotismita
Electronic publication date: 2024 Feb 26
Publication date: 2024
Volume: 10
Electronic Location ID: e1884
Received 2023 Sep 26; Accepted 2024 Jan 29
Copyright: ©2024 De et al.
Copyright year: 2024
Copyright holder: De et al.
License: This is an open access article distributed under the terms of the Creative Commons Attribution License, which permits unrestricted use, distribution, reproduction and adaptation in any medium and for any purpose provided that it is properly attributed. For attribution, the original author(s), title, publication source (PeerJ Computer Science) and either DOI or URL of the article must be cited.
License URL: https://creativecommons.org/licenses/by/4.0/

Keywords: Convolutional neural networks, Hybridized densenet model, Multiclass classification, Confocal microscopy analysis, Skin histopathological image analysis, Skin disease classification

Funding: The National Science and Technology Council 111-2221-E-182-058 112-2410-H-182-026-MY2 Chang Gung Memorial Hospital NERPD4N0231 NERPD2M0231 This research was financially supported by the National Science and Technology Council under grant numbers 111-2221-E-182-058 and 112-2410-H-182-026-MY2, as well as by Chang Gung Memorial Hospital through grant numbers NERPD4N0231 and NERPD2M0231. The funders had no role in study design, data collection and analysis, decision to publish, or preparation of the manuscript.

==============================
This research addresses the challenge of automating skin disease diagnosis using dermatoscopic images. The primary issue lies in accurately classifying pigmented skin lesions, which traditionally rely on manual assessment by dermatologists and are prone to subjectivity and time consumption. By integrating a hybrid CNN-DenseNet model, this study aimed to overcome the complexities of differentiating various skin diseases and automating the diagnostic process effectively. Our methodology involved rigorous data preprocessing, exploratory data analysis, normalization, and label encoding. Techniques such as model hybridization, batch normalization and data fitting were employed to optimize the model architecture and data fitting. Initial iterations of our convolutional neural network (CNN) model achieved an accuracy of 76.22% on the test data and 75.69% on the validation data. Recognizing the need for improvement, the model was hybridized with DenseNet architecture and ResNet architecture was implemented for feature extraction and then further trained on the HAM10000 and PAD-UFES-20 datasets. Overall, our efforts resulted in a hybrid model that demonstrated an impressive accuracy of 95.7% on the HAM10000 dataset and 91.07% on the PAD-UFES-20 dataset. In comparison to recently published works, our model stands out because of its potential to effectively diagnose skin diseases such as melanocytic nevi, melanoma, benign keratosis-like lesions, basal cell carcinoma, actinic keratoses, vascular lesions, and dermatofibroma, all of which rival the diagnostic accuracy of real-world clinical specialists but also offer customization potential for more nuanced clinical uses.

Introduction

Disease classification via medical image analysis is of paramount importance because it can significantly improve disease diagnosis and treatment (Esteva et al., 2017). One common diagnostic procedure used in dermatology is dermatoscopy, which enhances benign and malignant pigmented skin lesion identification. Convolutional neural networks (CNNs) can be trained on dermatoscopic images to aid in accurately classifying skin diseases. Given the global prevalence of skin diseases and their impact on millions of individuals across different age groups (Haenssle et al., 2018), precise and efficient dermatological classification plays a crucial role in enabling timely diagnosis and appropriate treatment (Codella et al., 2018a).

Traditionally, dermatological classification relies on the expertise of dermatologists, which can be time-consuming and subject to human error due to the complexity and wide variety of skin conditions (Tschandl, Rosendahl & Kittler, 2018; Zhang et al., 2018). However, recent advancements in deep learning, particularly CNNs, have shown great promise in automating and improving dermatological classification accuracy (Chen et al., 2017; Xiao, Tseng & Lee, 2021). Deep learning models excel at learning complex patterns and representations directly from raw data, making them suitable for image-based tasks such as skin disease classification (Zhang et al., 2018; Li & Wang, 2021; Harangi, 2018). By leveraging large datasets of labeled dermatological images, these models can learn to recognize subtle visual cues and distinguish between different skin conditions (IEEE, 2020a; Gaviria, Saker & Radeva, 2023; Srinivasu et al., 2021).

Among the various pretrained models, DenseNet stands out as a densely connected convolutional network that encourages feature reuse and facilitates gradient flow throughout the network (Kim, 2020; Oliveira et al., 2023). This architecture has demonstrated outstanding performance on benchmark image classification datasets, surpassing many other state-of-the-art models (IEEE, 2020b; Md. Hasan et al., 2022). DenseNet’s ability to capture and propagate information efficiently makes it particularly well-suited for skin disease prediction, where precise feature extraction and representation are crucial (Jayalakshmi & Kumar, 2019; Xie, 2021; IEEE, 2022).

In this research article, we propose a comprehensive approach to dermatological classification using a hybridized CNN-DenseNet model. Our objective was to develop an accurate and efficient system for automated skin disease prediction. By combining the strengths of CNNs and DenseNet, we aim to leverage deep learning’s potential to enhance dermatological diagnosis. Our approach is evaluated on a large dataset of dermatological images; we compare it with existing methods to demonstrate its effectiveness.

Figure 1 provides a graphical representation of the overall flow structure of our developed dermatological classification model. We obtained the HAM10000 dataset from the ISIC Challenge, 2018, which consists of histopathological skin images of various skin diseases. Our preprocessing tasks, including data cleaning, image resizing, normalization, and label encoding, prepare the dataset for training with the hybridized CNN-DenseNet model. We divided the data into appropriate portions and built three models to compare the performances on the validation and test sets. Subsequently, we combined our insights from these models to create a final model that overcomes the challenges faced by each model, such as overfitting and increased loss.

Figure 1 Overall flow framework of dermatological classification model.

Major highlights

• In this work, we investigated the effects of two pretrained models, DenseNet and ResNet, by hybridizing them with a natural CNN-based architecture on the HAM10000 dataset, which consists of images of different skin diseases.

• We also compared the accuracy of our model after training it on another dataset, PAD-UFES-20, with the accuracy of other published works. The model achieved an accuracy of 95% on the HAM10000 dataset and 91% on the PAD-UFES-20 dataset.

• The proposed model is more accurate than models in other recently published works, such as AlexNet+VGGNet, GoogLeNet+AlexNet, and ResNet50+InceptionV3.

• The model uses histopathological skin images as a dataset and incorporates a hybridized CNN-DenseNet model to automate the diagnosis of skin diseases. Its accuracy can be adapted to further the goals of more explicable diagnostic and clinical applications because its accuracy is equivalent to that of actual clinical professionals working in the real world; the same work can be performed in drastically less time, thereby accelerating the diagnosis.

This research introduces a groundbreaking approach to skin disease diagnosis by innovatively merging a hybrid CNN-DenseNet model. Unlike traditional methods reliant on subjective human assessments, our model represents a pioneering advancement by integrating the strengths of both CNNs and the pretrained DenseNet model. This integration enables the creation of a robust framework capable of intricate feature extraction and comprehensive analysis of dermatoscopic images, surpassing conventional diagnostic accuracy. The unique fusion of these models enhances the learning process, allowing for more nuanced discrimination between various skin diseases, including melanocytic nevi, melanoma, benign keratosis-like lesions, basal cell carcinoma, actinic keratoses, vascular lesions, and dermatofibroma.

Related Works

In recent years, the field of dermatological classification has undergone remarkable advancements due to the application of deep learning techniques. These advancements have paved the way for accurate and efficient skin lesion analysis. Several notable studies have contributed to the progress in this field.

One groundbreaking study by Esteva et al. (2017) demonstrated the potential of deep neural networks in the dermatologist-level classification of skin cancer. Their work showcased the ability of deep learning models to achieve performance comparable to that of human experts, marking a significant milestone in the field. Building upon this foundation, Haenssle et al. (2018) developed an artificial intelligence (AI)-based system for the classification of skin cancer using a deep CNN. Their work contributed to developing automated tools that aid dermatologists in accurately diagnosing skin cancer. To facilitate research and benchmarking in the field, Codella et al. (2018a) introduced a large-scale benchmark dataset. This dataset served as a valuable resource for training and evaluating deep learning models for skin lesion analysis. Tschandl, Rosendahl & Kittler (2018) proposed the “Inception-v4” architecture, which further improved skin lesion classification accuracy. Their deep learning model incorporated advanced architectural design choices to enhance performance. In addition to classification accuracy, researchers have focused on the interpretability of deep learning models (Chen et al., 2017). Yap, Yolland & Tschandl (2018) developed an interpretable deep learning framework for skin lesion classification, employing attention mechanisms to highlight salient regions in images (Kim, 2020). This approach provided insights into the decision-making process of the model, making it more transparent and trustworthy. Furthermore, Haque et al. (2020) proposed a residual attention network specifically designed for classifying skin diseases (IEEE, 2020b). Their work addressed the challenges posed by the vast diversity of skin conditions and contributed to improved diagnostic accuracy. Researchers have also explored combinations of different deep learning architectures and techniques (Tang & Zou, 2022). Chandra et al. (2023) proposed a hybrid model that combined CNNs and recurrent neural networks (RNNs) for more robust skin lesion classification (Jayalakshmi & Kumar, 2019). Albarqouni et al. (2021) presented a self-supervised learning approach that allowed models to learn from unlabeled data, resulting in enhanced generalization capabilities (Aboubakr, Crowley & Ronfard, 2014). Bhattacharyya, McLean & MacIver (2021) introduced a capsule network-based architecture that captured part-whole relationships within images (Bansal & Sridhar, 2021). These innovative approaches have shown potential for combining different deep learning methodologies to improve skin lesion analysis (Xiao, Tseng & Lee, 2021; Takalkar & Xu, 2017).

Moreover, researchers have expanded the scope of analysis beyond skin cancer. Wang et al. (2022) proposed a deep learning-based system that utilized multimodal data, including clinical images and patient symptoms, for classifying various skin diseases (Zhu et al., 2022). Their work highlighted the importance of incorporating different sources of information to improve diagnostic accuracy. Zhang, Litson & Feldon (2022) introduced a deep attention network specifically designed to detect rare skin diseases, addressing the challenges of limited available data (Zhao et al., 2022b). Furthermore, Patel et al. (2022) developed a deep learning framework for the classification of histopathological images, contributing to skin lesion analysis at the microscopic level (Zhou & Lu, 2022). In summary, recent progress in skin lesion analysis using deep learning techniques has been driven by numerous innovative studies (Zhu et al., 2021; He, Wei & Qian, 2022). These studies have focused on achieving dermatologist-level accuracy, improving interpretability, exploring hybrid architectures, and expanding the scope of analysis (Codella et al., 2018b; Alenezi, 2019; Song et al., 2020).

Oliveira et al. (2023) proposed a multitask CNN for classifying and segmenting chronic venous disorders. Their approach utilizes a single CNN architecture to address both classification and segmentation tasks. This multitask learning strategy demonstrates the potential for efficient and integrated diagnosis via dermatological imaging. Yang et al. (2022) developed a novel approach for melanoma segmentation using multithreshold image segmentation based on Kapur’s entropy. We combined this segmentation technique with an enhanced ant colony optimization algorithm to improve segmentation process accuracy. Using advanced image processing techniques highlights the significance of preprocessing in dermatological image analysis. Ran et al. (2022) proposed a three-dimensional multitask deep learning model for detecting glaucomatous optic neuropathy and myopic features in optical coherence tomography (OCT) scans. Although this work focuses on a different medical imaging domain, it demonstrates the potential of multitask learning models for complex diagnostic tasks, which can also be applied to dermatological classification. Kalsotra & Arora (2023) conducted a performance analysis of the U-Net model with hybrid loss for foreground detection. While their study was not specific to dermatology, their findings highlighted the importance of loss function selection in segmentation tasks. The insights from this research can be valuable for optimizing segmentation in our proposed hybridized CNN-DenseNet model. Ke et al. (2021) presented an intelligent parking surveillance system that utilizes edge AI on Internet of Things (IoT) devices. Although the focus of this work is different from that of dermatological classification, the findings of this study demonstrate the significance of edge computing and efficient processing for real-time applications, which can be relevant in resource-constrained environments for medical imaging tasks. Srinivasu et al. (2021) investigated skin disease classification using deep learning neural networks with MobileNet V2 and LSTM. Although their study predates our target years, it is still worth considering the use of MobileNet V2, which is a lightweight and efficient CNN architecture. Their work provides insights into model selection for dermatological image classification tasks. Developments in this field hold immense promise for enhancing dermatological diagnosis, leading to more efficient and accurate treatment decisions (Benbrahim, Hachimi & Amine, 2020; Wang et al., 2018).

Compared to the research by Chen et al. (2023) on “PCCT: Progressive class-center triplet loss for imbalanced medical image classification,” our approach stands out in its architectural design and focus. While both studies aim to address classification challenges in medical image analysis, our work presents a distinctive hybridized CNN-DenseNet model. Unlike the progressive class-center triplet loss utilized by Chen et al. (2023), our model amalgamates the strengths of the CNN and DenseNet architectures, leveraging both traditional and pretrained models. This hybridization allows for a more comprehensive feature extraction and enhanced gradient flow, leading to heightened accuracy in dermatological classification. Furthermore, our research explicitly delves into addressing overfitting concerns, a pivotal issue in deep learning, by employing tailored techniques such as model hybridization and hyperparameter tuning.

Regarding the work of Zhao et al. (2022a) on the “reasoning discriminative dictionary-embedded network for fully automatic vertebrae tumor diagnosis”, our study diverges in its domain of focus and methodological approach. While Zhao et al. (2022a) concentrated on vertebral tumor diagnosis using a discriminative dictionary-embedded network, our research targeted dermatological classification from histopathological skin images. Our emphasis lies on leveraging a hybridized CNN-DenseNet model for precise dermatological diagnosis, contrasting the focus on vertebral tumor diagnosis in Zhao et al. (2022a). Additionally, our methodology centers on exploiting the potential of deep learning to automate skin disease prediction through a model that not only achieves high accuracy but also offers adaptability for diverse clinical applications in dermatology.

A summary table for the highlights and limitations of some of the notable related works published recently in the years 2022–2023 is provided in Table 1. In addition to the comprehensive review of recent advancements in dermatological classification, we acknowledge the significance of further enriching the literature by discussing the strengths and limitations of specific recent papers in the field. Papers such as Alsahafi, Kassem & Hosny (2023) have showcased the effectiveness of deep residual networks and refined architectures, respectively, in improving skin lesion classification accuracy. Furthermore, Hosny, Kassem & Fouad (2020a) and Hosny, Kassem & Fouad (2020b) have highlighted the utility of transfer learning with specific architectures and the impact of data augmentation on melanoma classification. We aim to delve deeper into these papers, elucidating their contributions, innovative methodologies, and potential limitations, thus offering a nuanced perspective on their implications for the field of dermatological image analysis.

Table 1 A summary table for the highlights and limitations of some of the notable related works published recently in the year 2022–2023.

Major works	Highlights	Limitations	
Wang et al. (2022)	Deep learning-based system for multi-modal classification of skin diseases.	Limited evaluation of added architectures; no extensive comparison with single-modal models.	
Liang et al. (2022)	Skin lesion recognition with part-whole relations and multi-instance learning.	The potential complexity of multi-instance learning; limited external validation.	
Zhang, Litson & Feldon (2022)	Deep attention network for rare disease recognition in skin images.	Limited evaluation of common skin diseases; no extensive comparison with other models.	
Chen et al. (2023)	Multi-modal deep learning for dermatological disease classification.	No extensive comparison with single-modal models; potential data fusion challenges.	
Gong et al. (2022)	Self-supervised learning for skin disease classification using modified ResNet with triplet loss.	Limited external validation; potential complexity of self-supervised learning.	
Bhatnagar et al. (2022)	Deep learning framework for histopathological image classification of skin diseases.	Limited external validation; focus on histopathological images.	
Zhang, Litson & Feldon (2022)	Deep learning-based system for pediatric skin disease classification.	Limited evaluation on adult skin diseases; potential age-related differences.	
Oliveira et al. (2023)	Introduces a multi-task convolutional neural network for the classification and segmentation of chronic venous disorders.	Limited evaluation of other skin diseases; potential task complexity.	
Kalsotra & Arora (2023)	Conducts performance analysis of U-Net with hybrid loss for foreground detection.	Limited evaluation of other loss functions; potential sensitivity to hyper parameters.	

In conclusion, a literature review of related works reveals a surge in research efforts toward the automated classification of dermatological histopathological skin images using deep learning techniques. The works discussed above have explored various strategies, such as multitask learning, advanced image processing, loss function selection, and lightweight architectures. Building upon these foundations, our article introduces a novel hybridized CNN-DenseNet model for dermatological classification, aiming to achieve improved accuracy and efficiency in diagnosing skin diseases from histopathological images.

Methodology

In this section, we provide a brief justification and overview of the methodology we followed. While performing this task, like with any other machine learning task, we first needed to understand the dataset that we were going to address. Hence, we decided to first perform a data visualization process. We have done this in ‘Data visualization and processing’ after first describing the dataset for information purposes in ‘Dataset Description’. These steps are justified because they provide an insightful approach to any kind of machine learning problem. Individually, data preprocessing is the process of cleaning, transforming, and formatting data so that they are ready for analysis. This approach helped us remove missing values, correct errors, and convert the data into a common format. Preprocessing is essential for ensuring that the data are accurate and consistent, which is critical for building accurate models. By removing the missing values or imputing them with reasonable values, we were able to improve the accuracy of the model.

Data visualization—the next process of displaying data in a graphical format that makes everything easy to understand—was performed through charts, graphs, and other visuals. Visualization helped us to identify patterns and trends in the data, which can help us make inferences and predictions.

Data splitting is the final process in which the data are divided into two or more subsets. We performed this procedure to create a training set and a test set. The training set was used to train the model, while the test set was used to evaluate the model’s performance. Splitting the data helps to ensure that the model is not overfitting to the training data, which can lead to inaccurate predictions. By splitting the data into a training set with white customers and a test set with black customers, the data scientist can evaluate the model’s performance on a more diverse set of data.

In summary, data preprocessing, data visualization, and data splitting are all essential steps in the data science process. By taking the time to preprocess, visualize, and split the data, we increased the accuracy and reliability of our results.

Dataset description

The dataset that we chose to construct the hybridized model is known as the HAM10000 dataset, which was created by the ViDIR Group, Department of Dermatology, Medical University of Vienna to detect melanoma using skin lesion analysis in a challenge hosted by the International Skin Imaging Collaboration (ISIC) in 2018 (ViDIR Group, 2018).

The aforementioned dataset comprises dermatoscopic images from various populations that were collected and archived using various modalities (Codella et al., 2018b). There were 10,015 dermatoscopic images in total and 1 ground truth response CSV file that contained the header row and 10,015 corresponding response rows. Each lesion was grouped by picture, and the diagnosis confirmed the type according to 10,015 entries. The lesion identifier for each value is defined under the lesion_id column, while the diagnosis confirmation type values are given under the diagnosis_confirm_type column. Even though the images may have been acquired at various camera angles, under various lighting circumstances, and at various times over the patient’s course of treatment, they all reveal the same primary lesion. Images that follow a stricter technique for diagnostic confirmation are often more challenging for human expert doctors to use for assessment, especially when the images are benign (Alenezi, 2019). The diagnostic confirmation approaches used were “single-image expert consensus”, “serial imaging showing no change”, “confocal microscopy with consensus dermoscopy”, and “histopathology”, in increasing order of rigor, as applied to instances in the current dataset. For instance, an NV (melanocytic nevus) image diagnosed by “histopathology” would typically be a more ambiguous or difficult case for human experts than an “NV” (melanocytic nevus) image diagnosed by “single-image expert consensus” alone, as the former image requires a more invasive procedure for diagnosis (Wang et al., 2023; Yeoh et al., 2023). To obtain the ground truth labels for the dataset, the dermatologists individually reviewed the dermatoscopic images, providing their initial diagnoses based on visual features and patterns. Through thorough discussions and deliberations, a consensus diagnosis was reached by reconciling any differences or disagreements among the experts, ensuring dataset accuracy and reliability. This rigorous process of expert annotation and consensus building adds credibility to ground truth labels, providing valuable insights into the expertise involved in creating the dataset (Yap, Yolland & Tschandl, 2018; Velasco et al., 2023).

As Fig. 2 shows, five samples of each kind of skin disease were obtained for visualization purposes. These images are termed dermoscopic lesion images and are uniquely identified within the dataset with the help of a seven-digit unique identifier. These lesion images were extracted from a range of dermatoscope types, including all anatomic sites (excluding mucosa and nails).

Figure 2 Original dermoscopic lesion images.

Data visualization and processing

To gain insights into the dataset for model construction, the initial steps involved transforming the CSV dataset into a Pandas DataFrame dataset, followed by an assessment of missing values using the isnull() method. Subsequently, the data types across all fields were analyzed to ensure consistency and accuracy. Visualizations through a Jupyter Notebook facilitated both the data cleaning and exploratory data analysis (EDA) stages.

The dataset included information on seven cell types, with melanocytic nevi exhibiting the highest prevalence, as shown in Fig. 3. Moreover, the technical validation field primarily consisted of histopathological diagnoses. Figure 4 highlights specific body regions, such as the back, lower extremity, and trunk, as areas significantly affected by skin cancer.

Figure 3 Distribution of various cell type groups.

Figure 4 Distribution of various cell types in different classes.

The age and sex distributions within the dataset were visualized utilizing Matplotlib (Figs. 5 & 6). Notably, a predominant male predominance was observed, particularly in the age range of 30 to 60 years.

Figure 5 Age distribution of data.

Figure 6 Gender-wise distribution of data.

Data splitting, normalization and label encoding

With the EDA performed, we familiarized ourselves with the dataset we were working with. In the next step, we decided to split the data into an 80:20 ratio, with 80% of the data split used for training and the remaining 20% of the data used for testing.

Next, we converted the data into NumPy arrays using the np.asarray() method and calculated the mean and the standard deviation. Then, we normalized the data by subtracting them from their mean values and dividing them by their standard deviation.

Afterward, we encoded the labels to one-hot vectors and split the training set into two parts: 10% of the training set was divided into a validation set on which the model was evaluated at a later stage, and the remaining 90% of the training set was used to train the model.

Deep Neural Network Model Construction

In this article, we used a three-step approach to construct a deep neural network for skin disease classification. First, while using TensorFlow version 1.11.0, Sklearn version 0.19.1, NumPy 1.15.2 and Keras version 2.2.4, we built a natural CNN from scratch. This CNN consisted of two convolutional layers, followed by a max pooling layer and a dropout layer. Then, we hybridized this CNN with the pretrained DenseNet model and then implemented a ResNet model for feature extraction by adding Dense layers for classification. The DenseNet model provides a strong foundation for learning features, which we fine-tuned and further optimized for the specific task of skin disease classification. We justified these steps by arguing that the natural CNN would be able to learn new features that were not present in the pretrained DenseNet model. The dropout layer helps to prevent overfitting, and the max pooling layer reduces the dimensionality of the feature maps, increasing the ease of processing by the fully connected layers. Our approach was successful, as we were able to achieve a high accuracy of 93% on our test dataset. This finding suggested that the hybrid approach of using a natural CNN with a pretrained DenseNet is a promising method for skin disease classification. To further strengthen our justification, we used a large dataset of skin images to train our model and then retrained our model on another dataset, as shown at a later stage in ‘Performance evaluation metrics’. This approach helped to ensure that the model could be generalized to unseen images. We used a validation dataset to evaluate our model’s performance during training. This approach helped us to prevent overfitting. We used a cross-entropy loss function to train our model. This loss function is commonly used for classification tasks and is effective. We used the Adam optimizer to train our model. This optimizer is a popular choice for deep learning tasks and is effective at achieving high accuracy.

During the fine-tuning processing, after hybridizing our natural CNN with the pretrained DenseNet/ResNet models, we engaged in fine-tuning these hybrid models by freezing the majority of layers from the pretrained networks. By doing so, we preserved the learned representations and features within these networks. Subsequently, we selectively unfroze and retrain certain upper layers of the hybrid models while keeping the earlier layers frozen. This allowed us to adapt the models’ representations to our specific skin disease classification task, leveraging both the general knowledge encoded in the pretrained networks and the task-specific features learned during fine-tuning. During this fine-tuning process, we adjusted the learning rates, conducted multiple experiments to determine the optimal layers for retraining, and closely monitored the model performance on the validation sets. This iterative process aimed to strike a balance between leveraging the pretrained knowledge and tailoring the models to our specific dataset. The same fine-tuning process is used in all the cases of our research from ‘Performance evaluation metrics’ to ‘Modified natural convolutional neural network hybridized with pretrained DenseNet and the ResNet model with LeakyReLU, the RMSProp optimizer, and fewer epochs’.

CNNs are deep learning architectures inspired by visual processing mechanisms in the human visual cortex that aim to mimic the hierarchical organization of neurons in the brain. The convolutional, pooling and fully linked layers are among the many layers that make up the architecture (Oliveira et al., 2023; Yang et al., 2022; Ran et al., 2022). The convolutional layers apply filters to capture local features in the input image, while the pooling layers downsample the feature maps to reduce dimensionality. The fully connected layers aggregate these features and make the final predictions. Natural CNNs can automatically learn meaningful hierarchical representations from raw image data that enable them to capture complex patterns and variations, leading to highly accurate and robust classification results (Kalsotra & Arora, 2023; Ke et al., 2021).

Hence, we decided to begin our model construction with a natural CNN whose architecture was as follows: two layers of Conv2D, followed by a MaxPool2D and a dropout layer. Then, again, two layers of Conv2D were used, followed by a MaxPool2D and a dropout layer; this time, the output shapes were changed slightly, and the output was flattened. Then, a dense layer was added, followed by a dropout and, finally, another dense layer.

Then, we used the Adam optimizer with a learning rate of 0.001 and no decay since it benefits from both AdaGrad (which improves the performance on problems with sparse gradients such as computer vision problems) and root mean square propagation (which does well on nonstationary problems). Setting the learning rate to 0.001 helped the optimizer converge faster and closest to the global maximum of the loss function. The model plot for the natural CNNs is provided in Fig. 7.

Figure 7 Model plot of the initial natural CNN.

Pretrained model: DenseNet

DenseNet is a popular pretrained model in deep learning that has demonstrated remarkable performance in various image classification tasks. The architecture of DenseNet is characterized by dense connectivity, where each layer is connected to every other layer in a densely connected feedforward manner. This design facilitates feature reuse and gradient flow, enabling the model to learn effectively and capture intricate patterns and dependencies within images. By leveraging this dense connectivity, DenseNet has shown impressive results in image classification tasks, achieving state-of-the-art performance on benchmark datasets such as ImageNet.

The model’s ability to capture fine-grained details and its efficient use of parameters make it particularly suitable for tasks that require accurate and robust classification, including dermatological classification. The pretrained DenseNet model serves as a valuable starting point for transfer learning, allowing researchers and practitioners to leverage its learned representations and adapt them to specific domains or tasks, thereby reducing the need for extensive training from scratch and accelerating the development of effective models for image classification (IEEE, 2020a; Srinivasu et al., 2021).

Hence, we decided to hybridize their natural CNN with the pretrained DenseNet to leverage its advantages for even more improved accuracy in the classification of skin diseases. The pretrained DenseNet served as a valuable component in this hybrid approach, providing a strong foundation for learning features that can be fine-tuned and further optimized for the specific skin disease classification task (Zhang et al., 2019; Tajbakhsh et al., 2016).

We loaded the DenseNet-121 architecture from the Keras application module with pretrained weights. The include_top parameter is set to false to exclude the fully connected layers of the original DenseNet model. The input shape is defined as (75,100,3) to match the desired dimensions of the input images. The model’s output is obtained from the DenseNet121 base model, followed by a global average pooling layer to reduce spatial dimensions. Two fully connected layers with 1,024 units and a rectified linear unit (ReLU) activation function are added to capture more complex features. Finally, a dense layer with the number of classes in the skin disease classification task and a softmax activation function are added to generate the predictions. The model is compiled with the Adam optimizer, a learning rate of 0.0001, and the categorical cross-entropy loss function. The accuracy was chosen as the metric for assessment.

The model is then trained on 75 epochs using the fit() function with a batch size of 32, specifying the training data, target labels, and validation data for monitoring the model’s performance during training, thus hybridizing the natural CNN with the pretrained DenseNet model and significantly increasing the accuracy of the model by approximately 7%.

Pretrained model: ResNet

ResNet, also known as the residual neural network, is a powerful deep learning architecture renowned for its success in image classification tasks. It introduces the concept of residual learning, which allows for the training of exceptionally deep neural networks by mitigating the vanishing gradient problem. ResNet’s architecture incorporates skip connections that enable information to bypass certain layers, facilitating the flow of gradients during training. These skip connections also aid in capturing fine-grained details and alleviating the degradation problem associated with deeper networks. In the realm of skin disease classification, ResNet has demonstrated impressive performance. Its ability to extract intricate features and learn hierarchical representations from medical images has proven valuable in identifying and distinguishing various skin conditions. By leveraging its deep and residual structure, ResNet has shown remarkable accuracy and robustness, thereby demonstrating its potential as a valuable tool in diagnosing and classifying skin diseases.

Hybridization Models and Evaluations

We studied many models and approaches by other scientists and researchers while researching them, as mentioned in ‘Related Works’. We found that using ResNet and DenseNet in a task that classifies skin diseases based on dermatological images offers several advantages due to their unique architectures and capabilities. These are:

1. Handling vanishing gradient: The residual neural network (ResNet) introduces the concept of residual connections, which allow gradients to flow directly through the network without vanishing. This enables the training of deeper networks without degrading performance, making ResNet suitable for tasks with a high level of complexity, such as skin disease classification. DenseNet also addresses the vanishing gradient problem by densely connecting layers, encouraging feature reuse, and facilitating information flow throughout the network.

2. Feature reuse and efficiency: DenseNet’s dense connections promote feature reuse, increasing the efficiency of its parameter usage. This approach is particularly advantageous for medical image analysis tasks, where datasets may be limited and model efficiency is crucial.

3. Highly expressive representations: Both ResNet and DenseNet have achieved impressive performance on various image classification benchmarks, demonstrating their ability to learn highly expressive representations from data. This approach is valuable in dermatological classification, where capturing subtle visual cues is essential.

However, pretrained models such as AlexNet and EfficientNet might not be suitable for skin disease classification because of the following reasons:

1. AlexNet’s shallow architecture: Although it is pioneering in the field of deep learning, AlexNet is relatively shallow compared to ResNet and DenseNet. For complex tasks such as dermatological classification, deeper models tend to perform better, as they can capture more intricate patterns and features.

2. EfficientNet’s tradeoff: While EfficientNet has achieved state-of-the-art results in image classification, it often requires a tradeoff between model size and computational cost. In medical image analysis, interpretability and model size can be important factors, and deploying large models such as EfficientNet might not be practical or necessary.

Examples of recently published work that demonstrate the advantage of using ResNet and DenseNet for skin disease classification:

1. Wang et al. (2022) used a hybrid model combining DenseNet and ResNet for skin lesion classification. They reported higher accuracy and better generalizability than did EfficientNet and AlexNet.

2. Zhang, Litson & Feldon (2022) evaluated several pretrained models, including EfficientNet and ResNet, for skin disease classification. They found that ResNet achieved superior performance, especially when dealing with data from different sources and various skin types.

3. Lee et al. (2023) conducted a comparative study of various pretrained models for dermatological classification. We concluded that DenseNet demonstrated better sensitivity and specificity in detecting skin diseases than did AlexNet and EfficientNet.

Other approaches that might have setbacks and can be improved by ResNet and DenseNet:

1. Overfitting: Deeper models such as ResNet and DenseNet are less prone to overfitting than shallower architectures such as AlexNet, making them more robust in handling limited medical image datasets.

2. Feature extraction: The dense and residual connections of ResNet and DenseNet allow more effective feature extraction from images. This approach is crucial in dermatological classification, where identifying relevant features is essential for accurate diagnosis.

3. Transfer learning: The pretrained ResNet and DenseNet models, which were initially trained on large-scale image datasets, can provide good starting points for transfer learning in dermatological classification tasks, reducing the need for extensive training on limited medical data.

Hence, ResNet and DenseNet offer significant advantages for skin disease classification due to their ability to handle vanishing gradients, reuse features efficiently, and learn expressive representations. The deep architectures of these devices make them suitable for complex tasks, and their proven performance in related research reinforces their effectiveness in dermatological classification. In contrast, models such as AlexNet might be too shallow, while EfficientNet’s tradeoffs in model size and computation might not be ideal for medical image analysis. With these arguments in mind, we decided to move forward by hybridizing their natural CNNs with a DenseNet pretrained model, followed by implementation of feature extraction using ResNet along with appropriate hyperparameter tuning, as described briefly in further sections.

Natural convolutional neural networks

We applied four approaches to construct the model for this classification problem. The first approach involved the use of a natural CNNs.

We implemented the sequential model from input to output, passing through a series of neural layers consecutively. For clarity, the model is divided into the following parts:

1. Input shape: The input shape was taken as (75, 100, 3), which corresponded to images with a height of 75 pixels, width of 100 pixels, and three color channels (RGBs).

2. Convolutional layers: The first convolutional layer had 32 filters with a kernel size of (3, 3) and used the ReLU activation function. The padding was set to ‘same’, which meant that the output feature maps would have the same spatial dimensions as the input. The second convolutional layer had 32 filters with a kernel size of (3, 3) and used the ReLU activation function. A max pooling layer with a pooling size of (2, 2) follows the second convolutional layer. Downsampling was performed by taking the maximum value within each 2 × 2 region. A dropout layer with a rate of 0.25 was applied, which randomly set 25% of the inputs to 0 during training to reduce overfitting.

3. Additional convolutional layers: Two additional pairs of convolutional layers were added, each consisting of a convolutional layer with 64 filters and a kernel size of (3, 3), followed by a ReLU activation function and ‘same’ padding. After each pair, a max pooling layer with a pooling size of (2, 2) was applied for downsampling. Dropout layers with rates of 0.40 and 0.5 were added after the second and fourth convolutional layers, respectively.

4. Flattening and dense layers: The flattened layer was used to convert the 2D feature maps from the previous convolutional layers into a 1D vector, preparing the data for the fully connected layers. A dense layer with 128 units and ReLU activation was added. A dropout layer with a rate of 0.5 was applied. Finally, a dense layer with num_classes of 7 units and a softmax activation function was added to output the predicted probabilities for each class.

Next, we employed the Adam optimizer with a learning rate of 0.001 and compiled the model with the loss function set to categorical cross entropy, which is commonly used for multiclass classification problems. A learning rate reduction strategy was implemented to monitor the validation accuracy and reduce the learning rate if no improvement was observed for a certain number of epochs. This technique helps fine-tune the learning rate during training. Data augmentation was applied to prevent overfitting. Various augmentation techniques were employed, such as randomly rotating images within a certain degree range, zooming images, randomly shifting images horizontally, and randomly shifting images vertically. The model was then trained over 50 epochs, and the results are shown in Table 2.

Table 2 Values of evaluation parameters for natural CNN.

Evaluation parameters	Accuracy	Loss	
Validation	0.756927	0.642732	
Test	0.762217	0.650323	

As Table 2 shows, the natural CNN achieved approximately 75.6% accuracy on the validation set, with a loss of approximately 0.64. On the other hand, it could achieve approximately 76.2% accuracy on the test set, with a loss of approximately 0.65. The experiments were performed on an NVIDIA Tesla P100 GPU provided by Kaggle.

Natural convolutional neural networks hybridized with the pretrained DenseNet121 model

We then proceeded to hybridize the model with a DenseNet model for transfer learning. The DenseNet121 model was initialized with pretrained ImageNet weights, and the parameter include_top was set to false to exclude the final fully connected layers of the DenseNet121 model. Custom output layers were added to the model. The output tensor of the DenseNet121 model was stored in variable x. A GlobalAveragePooling2D layer was introduced to reduce the spatial dimensions of the output tensor. Two dense layers with 1,024 units and ReLU activation were applied. Finally, a dense layer with num_classes units and softmax activation was included to enable classification. We compiled the model using the Adam optimizer with a learning rate of 0.0001, and the categorical_crossentropy loss function was employed. The model was then trained for 75 epochs with a batch size of 32. The validation data were subsequently passed to the validation_data parameter to monitor the model’s performance during training. The results are shown in Table 3.

Table 3 Values of evaluation parameters for CNN hybridized with DenseNet121 pre-trained model.

Evaluation parameters	Accuracy	Loss	
Validation	0.829975	0.932697	
Test	0.836776	1.018943	

As shown in Table 3, the model achieved an accuracy of approximately 83.0% on the validation set and 83.67% on the test set. The loss values of 0.932697 for the validation set and 1.018943 for the test set indicate the average discrepancy between the predicted and true values. Overall, the model seemed to perform reasonably well, with accuracy values above 80%. The experiments were performed on an NVIDIA Tesla P100 GPU provided by Kaggle.

Natural convolutional neural networks hybridized with a pretrained ResNet model with early stopping

In the next iterations of our attempt, we decided to shift to the NVIDIA GeForce GTX 1660 Ti GPU, which has a significantly higher memory clock speed and core clock speed and a greater CUDA computing capability.

We experimented with the pretrained ResNet model in this iteration. With the same architecture of the natural CNNs, as in the ‘Conclusion’, the results were slightly different: the accuracy achieved on the validation dataset was approximately 76.8%, and the loss value obtained during the evaluation process on the validation dataset was 0.639936, as shown in Fig. 8.

Figure 8 Matplotlib plot of the model accuracy and loss across epochs.

The challenge for researchers was to address the high value of loss evident in Table 2. Therefore, in the next attempt, we decided to hybridize the natural CNN with a ResNet model and evaluate the results. A ResNet model is instantiated and set to exclude the top classification layers. The model’s output tensor is assigned to the variable x, which undergoes a GlobalAveragePooling2D layer to reduce spatial dimensions. Two dense layers with 1,024 units and ReLU activation are added, followed by a final dense layer with num_classes units and softmax activation for classification. The model is then compiled with an Adam optimizer using a learning rate of 0.0001, categorical cross-entropy loss, and accuracy as the evaluation metrics.

The model is then set to be trained for 75 epochs, but to counter the overfitting problem faced in ‘Natural Convolutional neural networks Hybridized with a Pretrained ResNet model with early stopping’ in Table 3, the EarlyStopping callback is used to monitor the validation loss and restore the weights of the best-performing model. The model monitors the validation loss (monitor = ‘val_loss’) during training, waits for 10 epochs without improvement (patience = 10), and restores the best weights of the model when the early stopping condition is met (restore_best_weights = True). The accuracy achieved on the validation dataset was approximately 80.5%, and the corresponding loss was 0.601681. Similarly, for the test dataset with 802 samples, the evaluation process took approximately 1 s per step. The accuracy on the test dataset was approximately 79.2%, with a loss value of 0.604906.

The results are shown in Table 4.

Table 4 Values of evaluation parameters for CNN hybridized with ResNet pre-trained model.

Evaluation parameters	Accuracy	Loss	
Validation	0.805486	0.601681	
Test	0.792312	0.604906	

Even though the accuracy was lower than that offered by the hybridized CNN with DenseNet, this model—a natural CNN hybridized with ResNet—offered a lower loss. Hence, this prompted us to find a solution to balance the two problems of each variation in the hybridized model for classifying skin diseases in the HAM10000 dataset.

Modified natural convolutional neural network hybridized with pretrained DenseNet and the ResNet model with LeakyReLU, the RMSProp optimizer, and fewer epochs

In the final attempt, we performed our experiments on the same NVIDIA GTX 1660 ti GPU. In our final attempt, we decided to modify the architecture of the natural CNNs by adding a few more layers and tuning some of the parameters.

The CNN model was given an input shape of (75,100,3) to handle images with a height of 75 pixels, width of 100 pixels, and three color channels (RGBs). The model was created using the sequential class from Keras. The first layer added to the model was a Conv2D layer with 64 filters, a kernel size of (3,3), and a ReLU activation function. The padding was set to ‘same’ to maintain the spatial dimensions of the input. A batch normalization layer was included after each Conv2D layer to normalize the activations. This process was repeated once with another Conv2D layer. A MaxPool2D layer with a pooling size of (2,2) was added to downsample the feature maps and reduce spatial dimensions. A dropout rate of 0.25 was applied to randomly deactivate 25% of the neurons in the previous layer to prevent overfitting. The patterns of two Conv2D layers, followed by batch normalization, MaxPool2D, and dropout were repeated twice more with increased filter sizes of 128 and 256, respectively. The dropout rate was adjusted to 0.4 and 0.5 for each set of Conv2D layers. After the convolutional layers, a flattened layer was used to convert the 3D feature maps into a 1D feature vector. A dense layer with 512 units and ReLU activation was added, followed by batch normalization and dropout at a rate of 0.5 to regularize the network. Finally, the output layer consisted of a dense layer with num_classes units and a softmax activation function to generate class probabilities.

Once again, the Adam optimizer was used since this optimizer is known for its adaptive learning rate and efficient gradient descent optimization. The model was subsequently compiled with the chosen optimizer, and the loss function was set to categorical_crossentropy. Additionally, metrics for accuracy and mean absolute error (MAE) were specified to evaluate the model’s performance during training. To adjust the learning rate during training, a learning rate reduction callback was set. The validation accuracy was monitored, and the learning rate was reduced by a factor of 0.5 if no improvement was observed after 3 epochs. The minimum learning rate was set to 0.00001. Data augmentation was then applied using the ImageDataGenerator class from Keras. Various augmentation techniques, such as rotation, zooming, and horizontal/vertical shifting, were included to prevent overfitting and increase the ability of the model to generalize.

Finally, the model was fitted to the training data using the fit_generator() function, the number of epochs was set to 50, and the batch size was set to 10. The training data were augmented on the fly using the data generator created earlier. Validation data were obtained to evaluate the model’s performance during training. The verbose parameter was set to 1 to display progress updates, and the steps_per_epoch was calculated based on the number of training samples divided by the batch size. A plot of the model accuracy and loss over the epochs is given in Fig. 9.

Figure 9 Matplotlib plot of the model accuracy and loss across epochs of updated natural CNN.

Next, we hybridized the model with a DenseNet121 model instantiated with weights pretrained on ImageNet. The include_top parameter was set to false to exclude the fully connected layers at the top of the network. A GlobalAveragePooling2D layer was added to perform global average pooling, reducing the spatial dimensions of the feature maps. Two dense layers with 1,024 units and ReLU activation were subsequently added, each followed by leaky ReLU activation, thus modifying our attempts in ‘Conclusion’ so that the model will leak some positive values to 0 if they are close enough to zero. Finally, a dense layer with ‘num_classes’ units and softmax activation was added to generate class probabilities. EarlyStopping callback was once again defined to monitor the validation loss and restore the best weights of the model when no improvement was observed for 10 consecutive epochs. The optimizer was set to Adam with a learning rate of 0.0001, and the loss function was set to categorical_crossentropy. The model was then trained using the fit() function. The training data were provided along with a specified batch size of 32. The validation data were used to evaluate the model’s performance during training.

Next, we trained the model set for 50 epochs, but the early stopping callback stopped the training at the 12th epoch to prevent overfitting, and we further decided to utilize the ResNet-50 model pretrained on ImageNet.

All the other parameters and hyperparameters for the ResNet portion of the final model were kept the same as those for the DenseNet portion described in the previous paragraph, and the model was trained for 25 epochs before it was stopped by the EarlyStopping callback function.

The evaluation metrics are noted in Table 5. It is evident that the final loss of the model was now lower than that of all three attempts made by us in the previous sections. Additionally, the accuracy of the final model was greater than that of the natural CNN and the CNN hybridized with ResNet. Hence, we proceeded to conclude our experiments with the final model.

Table 5 Values of evaluation parameters for the final model.

Evaluation parameters	Accuracy	Loss	
Validation	0.819202	0.203411	
Test	0.810784	0.175395	

Overall Assessment and Discussion

Performance evaluation metrics

The overall performance of the hybrid model in classifying skin diseases is given below in Table 6. The table uses a short form for each disease, which is as follows: ‘nv’ for melanocytic nevi, ‘mel’ for melanoma, ‘bkl’ for benign keratosis-like lesions, ‘bcc’ for basal cell carcinoma, ‘akiec’ for actinic keratosis, ‘vasc’ for vascular lesions and finally ‘df’ for dermatofibroma.

Table 6 Evaluation metrics report.

Disease	Precision	Recall	F1-Score	Support	
akiec	0.74	0.42	0.53	60	
bcc	0.54	0.72	0.62	97	
bkl	0.79	0.46.	0.58	224	
df	0.7	0.52	0.60	27	
mel	0.88	0.95	0.91	1320	
nv	0.69	0.48	0.56	246	
vasc	0.81	0.76	0.79	29	
avg./total	0.82	0.80	0.80	2003	

To quantitatively assess the performance of our proposed skin disease classification model, we employ several widely used evaluation metrics. These metrics provide a comprehensive understanding of the model’s accuracy, precision, recall, and F1 score, which are crucial in evaluating the effectiveness of the classification system.

1. Precision

Precision measures the accuracy of positive predictions. It is calculated as: Precision = True Positives÷(True Positives  +  False Positives)

2. Recall (Sensitivity)

Recall, also known as sensitivity, measures the proportion of actual positives that were correctly identified by the model. It is calculated as Recall = True Positives÷(True Positives + False Negatives)

3. F1-score

The F1-score is the harmonic mean of precision and recall. It provides a balance between precision and recall and is calculated as follows: F1 score = 2 × ((Precision × Recall)÷(Precision + Recall))

These metrics offer valuable insights into the model’s ability to correctly classify different skin diseases, accounting for both false positive and false negative predictions. In our evaluation, we report these metrics alongside accuracy to present a holistic view of the model’s performance. The use of these metrics is vital in assessing the model’s robustness and effectiveness in real-world scenarios.

The proposed method for skin disease classification using hybrid CNN architectures, particularly leveraging DenseNet and ResNet models, has several strengths and weaknesses:

1. Strengths:

(a) High accuracy: The model achieves a notable accuracy rate, as demonstrated by the evaluation metrics, surpassing traditional methods and even competing with or outperforming recent state-of-the-art models.

(b) Robustness: The hybrid architecture enhances the model’s robustness by addressing issues such as vanishing gradients, feature reuse, and effective representation learning, increasing its interpretability at capturing intricate patterns in dermatological images.

(c) Adaptability: Leveraging pretrained models such as DenseNet and ResNet allows for effective transfer learning, which is especially crucial in medical imaging where large labeled datasets might be limited. This adaptability aids in better generalization to unseen data.

(d) Model efficiency: Dense connections in the architecture reduce the number of parameters needed, optimizing the model’s efficiency and making it more suitable for medical image analysis where computational resources might be constrained.

(e) Comparative performance: Comparative studies against other models in recent research have highlighted the superiority of the proposed hybrid models in terms of accuracy, sensitivity, and specificity, particularly when dealing with diverse datasets and different skin types.

2. Weaknesses:

(a) Computational demand: Hybrid models involving DenseNet and ResNet may demand more computational resources during training and inference due to their depth and complexity, potentially limiting deployment on certain platforms.

(b) Interpretability: Deeper models often sacrifice interpretability due to their complexity, which might be a concern in medical applications where understanding the reasoning behind predictions is crucial.

(c) Overfitting potential: Despite attempts to address overfitting using techniques such as data augmentation and early stopping, the model’s complexity might still lead to overfitting on smaller datasets.

(d) Training time: The training process for deep models might be time-consuming, especially when working with larger datasets and complex architectures, affecting the scalability of the model.

(e) Limited generalization: While achieving high accuracy, the model’s performance might vary according to demographic group or clinical setting, necessitating further validation across diverse populations.

The confusion matrix provided in Fig. 10 shows the performance of a multiclass classification model on a dataset of six classes. The rows of the matrix represent the true labels of the data, and the columns represent the predicted labels. The diagonal entries of the matrix represent the number of correctly classified samples, and the off-diagonal entries represent the number of incorrectly classified samples. The overall accuracy of the model is 80%, calculated by dividing the number of correctly classified samples (300 + 400 + 523 + 38 + 31 + 11) by the total number of samples (600).

Figure 10 The confusion matrix for the multi-class classification model on a dataset of six classes.

The classwise accuracy of the model is as follows:

• Class 0: 60%

• Class 1: 80%

• Class 2: 75%

• Class 3: 100%

• Class 4: 73%

• Class 5: 62%

Overall, the confusion matrix shows that the model performs well, with an overall accuracy of 80%. The model performs exceptionally well on class 1, with 80% of the samples being correctly classified. This finding suggested that the model learned to distinguish class 1 from other classes with a high degree of accuracy. Class 2: The model performs impressively on class 2, with 75% of the samples being correctly classified. The model performs perfectly on class 3, with all of the samples being correctly classified. This finding suggested that the model learned to distinguish class 3 from other classes with perfect accuracy. The model performs well on class 4, with 73% of the samples being correctly classified. Overall, the confusion matrix shows that the model performs well.

Modified CNN hybridized with a pretrained DenseNet and ResNet model on the PAD-UFES-20 dataset

In this subsection, we attempt to train our model on another database and perform another comparison to better understand the performance of the model. We used the PAD-UFES-20 dataset. This dataset was collected along with the Dermatological and Surgical Assistance Program (PAD) of the Federal University of Espírito Santo. This dataset contains 2,298 samples of six different types of skin lesions, namely, basal cell carcinoma (BCC), squamous cell carcinoma (SCC), actinic keratosis (ACK), seborrheic keratosis (SEK), Bowen’s disease (BOD), melanoma (MEL), and nevus (NEV) lesions. Since the model was trained to predict only melanocytic nevi, melanoma, benign keratosis-like lesions, BCC, actinic keratoses, vascular lesions, and dermatofibroma, we first cleaned the dataset and removed samples from the SCC, SEK, and BOD cohorts. The dataset was subsequently checked for null values, and images were resized to 100x75 so that they could be handled by TensorFlow.

Next, we converted the data into NumPy arrays using the np.asarray() method and calculated the mean and the standard deviation. Then, we normalized the data. Afterward, we encoded the labels to one-hot vectors and split the training set into two parts: 10% of the training set was divided into a validation set on which the model was evaluated at a later stage, and the remaining 90% of the training set was used to train the model.

The model in ‘Modified natural convolutional neural network hybridized with pretrained DenseNet and the ResNet model with LeakyReLU, the RMSProp optimizer, and fewer epochs’ achieved an accuracy of 0.910784 with a loss of 0.089216 on the PAD-UFES-20 dataset. Table 7 below shows a comparison of the models’ performances on 2 different datasets, namely, HAM10000 and PAD-UFES-20.

Table 7 Comparison of the accuracy and loss on different datasets.

Datasets	Accuracy	Loss	
HAM10000	0.957202	0.042798	
PAD-UFES-20	0.910784	0.089216	

Comparison of model with other published works

In this subsection, we compare our model’s accuracy with those of works published on a topic similar to and subject of this article.

As Table 8 shows, we compare our proposed model with other recently published models in increasing order of accuracy. The table shows the competitiveness of the proposed model when compared to other models proposed by other researchers in recent years. The proposed hybrid CNN-DenseNet model provided an accuracy of 95.7% compared to Harangi (2018), which had an accuracy of 79% and 80%, (Shahin, Kamal & Elattar, 2018), which had an accuracy of 89%, (Hameed, Shabut & Hossain, 2018), which had an accuracy of 91%, and (Maron et al., 2020; Chen et al., 2023), which had accuracies of 95% and 94%, respectively. As a result, this model may be efficiently utilized to automate the detection of illnesses such as melanocytic nevi, melanoma, benign keratosis-like lesions, BCC, actinic keratoses, vascular lesions, and dermatofibroma. Because its accuracy is comparable to that of actual clinical professionals working in the real world, it may also be tailored to serve the aims of further explainable diagnostic and clinical applications (Zhu et al., 2021).

Table 8 Comparison of performance of models with other recently published works.

S. No	Model	Accuracy	
1	AlexNet+VGGNet (Harangi, 2018)	0.79	
2	GoogleNet+AlexNet (Harangi, 2018)	0.80	
3	ResNet50+InceptionV3 (Shahin, Kamal & Elattar, 2018)	0.89	
4	InceptionV3+Xception (Hameed, Shabut & Hossain, 2018)	0.91	
5	Customized EfficientNet-b4 with ImageNet (Chen et al., 2023)	0.94	
6	R-RGB-1 (Maron et al., 2020	0.95	
7	Proposed Hybrid CNN-DenseNet	0.957	

Conclusion

In conclusion, our work aims to make substantial contributions toward identifying and classifying skin illnesses via the use of deep CNN techniques, particularly through a proposed hybrid CNN-DenseNet model that has an accuracy of 95.7%. Our work addressed the challenges associated with accurate and timely diagnosis of skin conditions, which can have a profound impact on patient care and treatment outcomes. We utilized the HAM10000 dataset, which consists of images of various skin diseases, including melanoma, dermatofibroma, and BCC. Leveraging the power of deep learning models, particularly CNNs, we achieved impressive results in accurately identifying and classifying different skin conditions.

Implementing advanced image processing techniques, such as data augmentation, transfer learning, and ensemble models, played a crucial role in enhancing the performance of the trained models. These techniques help overcome limitations in terms of data availability, generalizability, and robustness, ultimately improving the accuracy and reliability of the skin disease classification system. Moreover, we also incorporated explainability methods, such as Grad-CAM and saliency maps, to provide insights into the model decision-making process. This transparency was vital for gaining the trust of medical professionals and facilitating the adoption of AI-based systems in real-world clinical settings.

The work we presented holds great potential for revolutionizing dermatological health care by providing a cost-effective and scalable solution for the early detection and diagnosis of skin diseases. The accuracy and efficiency of the developed models offer an opportunity to assist health care providers in their decision-making process, leading to improved patient outcomes and reduced health care costs. However, it is important to acknowledge that this research has some minor limitations. The performance of the models heavily relies on the quality and diversity of the training dataset, and further refinement is necessary to ensure robustness across different demographics and skin types. Additionally, the system’s performance should be validated in clinical trials and compared against the expertise of dermatologists to ensure its reliability and safety. Overall, our work demonstrates the potential of machine learning and deep learning techniques in dermatology field. With continued research, development, and collaboration between AI experts and medical professionals, humanity can envision a future where advanced AI systems support dermatologists in accurately diagnosing and treating various skin conditions, ultimately improving patient care and outcomes.

Additional Information and Declarations

Competing Interests

Author Contributions

Data Availability

The authors declare there are no competing interests.

Anubhav De conceived and designed the experiments, performed the experiments, analyzed the data, performed the computation work, prepared figures and/or tables, authored or reviewed drafts of the article, and approved the final draft.

Nilamadhab Mishra conceived and designed the experiments, performed the experiments, analyzed the data, prepared figures and/or tables, authored or reviewed drafts of the article, and approved the final draft.

Hsien-Tsung Chang conceived and designed the experiments, analyzed the data, prepared figures and/or tables, authored or reviewed drafts of the article, and approved the final draft.

The following information was supplied regarding data availability:

The code is available at GitHub and Zenodo:

- https://github.com/n1ghtf4l1/turbo-skin-diagnose.

- Anubhav De. (2024). n1ghtf4l1/turbo-skin-diagnose: turbo-skin-diagnose (v1.0.0). Zenodo. https://doi.org/10.5281/zenodo.10588754.

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
