# Peer review of "An approach to the dermatological classification of histopathological skin images using a hybridized CNN-DenseNet model"

_PeerJ Computer Science, doi:10.7717/peerj-cs.1884_

## Round 0.1 · original submission · Major Revisions

Based on the referee reports, I recommend a major revision of the manuscript. The author should improve the manuscript, taking carefully into account the comments of the reviewers in the reports, and resubmit the paper.

**Language Note:** The review process has identified that the English language must be improved. PeerJ can provide language editing services - please contact us at [email protected] for pricing (be sure to provide your manuscript number and title). Alternatively, you should make your own arrangements to improve the language quality and provide details in your response letter. – PeerJ Staff

Reviewer 1 ·

Basic reporting

This article is written in good English and do a good job in dermatological classification of histopathological skin images. It addresses challenges like overfitting, which is an important issue in deep learning. Some discussions of the related works could be strengthened. For example, please discuss the differences between your work and:
1. "K Chen, et.al. PCCT: Progressive class-center triplet loss for imbalanced medical image classification" 2. "S Zhao, et. al. Reasoning discriminative dictionary-embedded network for fully automatic vertebrae tumor diagnosis"

Experimental design

The authors carry out sufficient experiments.

Validity of the findings

The proposed method is validated by several datasets, which, in my opinion, meets the standards of PeerJ.

Additional comments

no comment

Reviewer 2 ·

Basic reporting

1. There are a few grammar or verbal problems that should be addressed:
(a) In the article, the authors sometimes use the third person to describe their work (e.g. the researcher/they in session 1, 3.2, and 3.3) but sometimes use the first person (e.g. we/our in the abstract and session 1 and 7). The authors also use the third person to describe previously published works. It makes readers hard to distinguish the object that the author is talking about. So I suggest the author use "we/our" to discuss their own work and use "they" to describe others' work.
(b) line 232, "To obtain the ground truth labels for the dataset the dermatologists individually reviewed the dermatoscopic images" A comma should be added to the sentences (...dataset, the...)

2. Some figures can be further improved:
(a) The words in Figure 2 are too small to be seen.
(b) The author does not print the title of the x-axis and y-axis for Figures 3, 4, 5, and 6.
(c) What is the meaning of 138437456055600 in Figure 7? The author does not clarify the meaning of this number.
(d) I suggest the author provide the model plot of CNN-RestNet and CNN-DenseNet, which can help reader understand the hybrid model better.

Experimental design

1. The author does not provide the version of tensorflow/keras and other dependent packages.
2. I cannot find the early stop criterion for CNN-DenseNet
3. The structures of CNN-DenseNet and CNN-ResNet are not clearly defined. From the paragraph of line 333 (session 4.2) and session 4.3, I can hardly understand how CNN and DenseNet/ResNet are connected. As the authors said, the densenet-121 is append by a global average pooling layer, two fully connected layers with 1024 units, a ReLU activation function, and a dense layer with the number (lines 333-337) but there is no description of the CNN here. There is also no description of the structure of CNN-ResNet in session 4.3. So I suggest authors to clarify these details. If authors can provide code files, that will be much better for reproducibility.
4. I cannot find the fine-tune detail of CNN-DenseNet/CNN-ResNet. The author just mention the word "fine-tune" in line 273 and 328 but they do not mention how they apply to CNN-DenseNet/CNN-ResNet

Validity of the findings

The author has compared the accuracy of their CNN-DenseNet model and others' models. However, these models are not trained by the same dataset or the data are not under the same preprocessing. So, the performance of these models is not comparable. The author should use their own dataset to train and test those models to make the accuracy comparable.

Additional comments

Predicting skin diseases from dermatoscopic images is a meaningful topic for human health. The author proposed a CNN-DenseNet model to accurately predict skin diseases. Compared to the existing models, it shows great improvement in this task. However, the authors did not describe the model clearly and the evaluation of their model is not sufficiently reliable. I suggest author address these issues to improve their works.

Reviewer 3 ·

Basic reporting

1. The abstract is very long. Please shorten it.
2. The abstract must summarize these points: what is the problem and challenge?
3. The paper needs to be revised by an English expert.
4. The paper needs to be revised by a dermatologist expert.
5. The author needs to shine on the novelty of the proposed method to be readable and more precise.
6. I advise authors to merge the subsection “Convolutional Neural Networks” to the main section “Deep Neural Network Model Construction.
7. The literature in the “Hybridization Models and Evaluations” section must be moved to related work, and I must talk about these limitations.
8. To enrich the literature, I encourage the authors to discuss the pros and cons of recent papers such as:
- Skin-Net: a novel deep residual network for skin lesions classification using multilevel feature extraction and cross-channel correlation with detection of outlier
Journal of Big Data, vol. 10 (1), Article No. 105, 2023.
- Refined Residual Deep Convolutional Network for Skin Lesion Classification
Journal of Digital Imaging, vol. 35 (2), pp. 258-280, 2022.
- Classification of skin lesions into seven classes using transfer learning with AlexNet
Journal of digital imaging, vol. 33, pp. 1325-1334, 2020.
- Skin melanoma classification using ROI and data augmentation with deep convolutional neural networks
Multimedia Tools and Applications, vol. 79, pp. 24029-24055, 2020.

9. The section “Data Visualization and Processing” is unclear and misunderstood.
10. What is meant by procure dataset?
11. Section 5 is divided into several subsections. I think this division is not meaningful.
12. The output of the pre-processing step must be clarified by skin image before and after each stage.
13. Used equations to measure the performance of the proposed method need to be added.
14. The symbols of all equations must be defended accurately.
15. What are the strengths and weaknesses of the proposed method?
16. A discussion of the results should be added.
17. A confusion matrix must added.
18. ROC must be added.
19. The conclusion is very long and needs to be summarized.

Experimental design

No comments

Validity of the findings

No comments

Additional comments

No comments

---

## Round 0.2 · Minor Revisions

Kindly revise the manuscript as per the reviewer suggestions and resubmit it.

Reviewer 1 ·

Basic reporting

The authors have dealed with my concerns. I recommend accept this article.

Experimental design

no comment

Validity of the findings

no comment

Additional comments

no comment

Reviewer 2 ·

Basic reporting

1. The resolution of the words in Figure 2 is still low.
2. The author still does not print the title of the x-axis and y-axis for Figures 3, 4, 5, and 6.
3. The author explains that the number 138437456055600 in Figure 7 is the number of nodes in the pipeline. What are the nodes? The 138437456055600 nodes as input of neural network? That's quite weird. Usually, the input of a neural network is the shape of the data rather than the "node".

Experimental design

Based on the code provided by author, the CNN-Resnet does not connect the CNN and Resnet:
https://www.kaggle.com/code/anubhavde/skinsentry-mark-v
"""
model = ResNet50(weights='imagenet', include_top=False, input_shape=(75, 100, 3))

x = model.output
x = GlobalAveragePooling2D()(x)
x = Dense(1024)(x)
x = LeakyReLU()(x)
x = Dense(1024)(x)
x = LeakyReLU()(x)
predictions = Dense(num_classes, activation='softmax')(x)
model = Model(inputs=model.input, outputs=predictions)
"""
The author only added some Dense layers for ResNet50 rather than CNNs. I suggest the author check the code to see whether there are errors here.

Validity of the findings

The issues have been addressed.

---

## Round 0.3 · accepted · Accept

The article is accepted for publication.

Reviewer 2 ·

Basic reporting

The issues have been addressed.

Experimental design

The issues have been addressed.

Validity of the findings

The issues have been addressed.